

# Quantifying uncertainty from aerosol and atmospheric parameters and their impact on climate sensitivity

Christopher G. Fletcher[1], Ben Kravitz[2], and Bakr Badawy[1,*]

[1]Department of Geography and Environmental Management, University of Waterloo, Waterloo ON, Canada
[2]Atmospheric Sciences and Global Change Division, Pacific Northwest National Laboratory, Richland, WA, USA
[*]now at: Environment and Climate Change Canada, Dorval, Québec, Canada

**Correspondence:** Christopher G. Fletcher, 200 University Ave. West, Waterloo, ON N2L 3G1, Canada.
(chris.fletcher@uwaterloo.ca)

**Abstract.** Climate sensitivity in Earth System Models (ESMs) is an emergent property that is affected by structural (missing or inaccurate model physics) and parametric (variations in model parameters) uncertainty. This work provides the first quantitative assessment of the role of compensation between uncertainties in aerosol forcing and atmospheric parameters, and their impact on the climate sensitivity of the Community Atmosphere Model, Version 4 (CAM4). Running the model with prescribed ocean

5 and ice conditions, we perturb four parameters related to sulfate and black carbon aerosol radiative forcing and distribution, as well as five atmospheric parameters related to clouds, convection, and radiative flux. The atmospheric parameters explain more than 85% of the variance in climate sensitivity for the ranges of parameters explored here, with two parameters being the most important: one controlling low cloud amount, and one controlling the timescale for deep convection. Although the aerosol parameters strongly affect aerosol optical depth, the effects of these aerosol parameters on climate sensitivity are

10 substantially weaker than the effects of the atmospheric parameters. Based on comparisons to inter-model spread of other ESMs, we conclude that structural uncertainties in this configuration of CAM4 likely contribute three times more to uncertainty in climate sensitivity than parametric uncertainty. We provide several parameter sets that could provide plausible (measured by a skill score) configurations of CAM4, but with different sulfate aerosol radiative forcing, black carbon radiative forcing, and climate sensitivity.

## 1   Introduction

Climate models are exceptional tools for understanding contributions of different forcing agents to climate change. When forced with observed climate forcings (e.g., greenhouse gases and aerosols), they are skillful at reproducing past climate change (Flato et al., 2013). Their ability to do so builds confidence that climate models can provide plausible projections of future climate change under various assumptions about ranges of changes in forcing agents (i.e., emissions scenarios).

20     At the core of these projections is the climate sensitivity (CS) of the model, which is an emergent property of a model that describes how much global warming will be simulated by the model for a prescribed change in the carbon dioxide ($CO_2$) concentration. Variations in CS arise due to *structural uncertainty* (how models represent the underlying physics of the climate system) and *parametric uncertainty* (how the simulation varies with model parameters, typically those associated with sub-



grid-scale parameterizations). Each parameter in a model parameterization has an associated uncertainty due to deficiencies in representing model physics with these parameterizations, or a lack of available observational data to constrain the physics. A third source of uncertainty comes from the emissions of climate forcings—particularly anthropogenic and natural aerosols—which are also uncertain to some degree, and have plausible ranges (e.g., Bond et al., 2013; Samset and Myhre, 2015; Ban-Weiss et al., 2011).

Reducing the uncertainty in CS is a key focus of climate science research. Fortunately, there are tools available to provide constraints on the estimates. Kiehl (2007), and more recently Forster et al. (2013), observed that there is a statistical relationship among climate models (the "Kiehl curve"), where models that are the most sensitive to $CO_2$ also tend to show largest historical radiative forcing by aerosols. As such, by exploring uncertainties in key model parameters and/or aerosol emissions rates, we expect to be able to alter a model's climate sensitivity, effectively moving the model along the Kiehl curve. With a sufficiently large number of such parameter sets, we can identify which versions of the model (i.e., sets of parameters or emissions rates) produce plausible climates. We hypothesize that, through this procedure, it may be possible to create multiple versions of a single climate model that plausibly reproduce observations but have discernibly different climate sensitivities.

The idea of obtaining multiple plausible versions of a model with different climate sensitivities was also explored by Mauritsen et al. (2012) for the Max Planck Institute for Meteorology's climate model. These authors found that the CS was not as sensitive to parameters as expected; however, they did not attempt to explore uncertainty in the forcing. Similarly, Golaz et al. (2013) altered the settings of a cloud detrainment parameter in the deep convection scheme, and produced three versions of the GFDL CM3 model with very different climate sensitivities, but again did not vary the forcing, and so two of the candidate versions were not able to reproduce the transient evolution of historical global mean surface air temperature. A recent, and highly relevant, contribution is that of Regayre et al. (2018), who show that uncertainties in historical aerosol radiative forcing, and top-of-atmosphere shortwave radiative flux, in a comprehensive chemistry-climate model are controlled by a combination of aerosol parameters *and emissions*, as well as uncertain atmospheric parameters. Their results show that, particularly in recent decades, constraining aerosol and atmospheric parameters allows regional climate impacts of aerosols to be more faithfully reproduced.

This area of research is closely related to Uncertainty Quantification (UQ), and to the more applied topic of model tuning (see also: Watanabe et al. (2010); Gent et al. (2011); Hourdin et al. (2016, 2013); Zhao M. et al. (2018)). Modern climate and Earth System Models (ESMs) are so complex, and with so many tunable parameters, that running comprehensive calibration schemes with the full dynamical model is impractical. The typical approach is to conduct one-at-a-time (OAT) tests, where a single uncertain parameter is varied while holding all other parameters at their default value (e.g., Covey et al., 2013). While a lot can be learned about a model in OAT mode, this approach neglects potentially important parameter interactions, which can only be studied in vastly more expensive all-at-a-time (AAT) sampling designs. A very useful approach, therefore, is to construct a statistical emulator trained on output from the dynamical model, which is used to predict unsampled outputs from the dynamical model. Many recent studies have used Gaussian Process (GP) emulators to sample an almost infinitely-large number of parameter combinations in climate and atmospheric models (Regayre et al., 2018; McNeall et al., 2016; Lee et al., 2011; Carslaw et al., 2013). The GP emulator is a Bayesian statistical technique that fits a smooth nonlinear function to a set of





training data based on some prior information/assumptions, and provides an estimate of its own posterior uncertainty for each prediction (Lee et al., 2011).

The main goal of this paper is to quantify how much variation in climate sensitivity within a single climate model (CAM4, the atmospheric component of CCSM4/CESM-CAM4; see Section 2.1 below) is a function of uncertainties in aerosol forcing and atmospheric parameters. We will use statistical emulation to test a very large ($O[10^6]$) number of parameter/forcing combinations, to identify those combinations that yield plausible climates, but with different climate sensitivities. In other words, we seek to quantify the degree of equifinality in CAM4's climate sensitivity. This work differs in focus from the recent study of Regayre et al. (2018), because here we explicitly assess the combined impact of parametric uncertainty and forcing uncertainty on climate sensitivity. While the conceptual idea of the Kiehl curve has existed for more than a decade, this work is, to our knowledge, the first explicit attempt to move a single climate model to different parts of the curve, and to quantify the impact on climate sensitivity. We stress that this contribution does not tackle the problem of calibration to observations: all simulations are conducted as perturbations to preindustrial (1850) conditions, and are compared to the default version of CAM4 as a reference. The idea is that we will end up with a series of High/Low CS candidate models that could be run with interactive ocean components, to produce transient RCP-type simulations (historical and future scenarios) beginning in the preindustrial era.

## 2 Data, Models and Methods

### 2.1 CAM4 model and atmospheric parameters

We use the National Center for Atmospheric Research (NCAR) Community Atmosphere Model Version 4 (CAM4), the atmospheric component of the Community Climate System Model Version 4 (CCSM4) and the Community Earth System Model Version 1.0.4 (CESM1), fully documented by Gent et al. (2011) and Collins et al. (2006). Hereafter, we refer to this model simply as "CAM4". In this study, CAM4 is run at coarse horizontal resolution (3.75° longitude × 3.75° latitude) with 26 vertical layers extending from the surface to 3 hPa (∼40 km). The coarse horizontal resolution is selected to increase computational efficiency, and is appropriate to represent the broad-scale features of the climate response described in this study (Shields et al., 2012). In addition, we note that this model configuration includes a crude representation of the stratosphere (only 4 layers are located above 100 hPa); however, since our primary focus is on radiation, and not circulation, we consider this resolution to be sufficient for the purpose of separating the radiative effects of aerosols in the troposphere and stratosphere. This is also the same vertical resolution that was employed in a similar manner by Ban-Weiss et al. (2011). All simulations are performed in preindustrial (1850) mode: the $CO_2$ concentration is fixed at 284 ppmv, while other atmospheric constituents are prescribed from a monthly-varying climatology. For natural and anthropogenic aerosols the climatology is taken from a simulation using interactive (i.e., prognostic) chemistry (Lamarque et al., 2010), and the output is taken as the mean of all realizations. The uncertainty due to atmospheric and oceanic initial conditions is estimated using the standard deviation of the realizations.

We assess climate sensitivity (CS) in CCSM4/CESM-CAM4 using the method proposed by Cess et al. (1991), where a uniform 2 K warming is applied to global sea surface temperatures (SSTs). The Cess CS is then given by $CS = \Delta F / \Delta T$,





where $\Delta T$ is the change in global annual mean near-surface air temperature, and $\Delta F$ is the change in global annual mean top-of-atmosphere net radiative flux ($\Delta F$). The changes are evaluated as the difference between the simulation with warmed SSTs minus the simulation with preindustrial SSTs. The CS diagnostic is useful for model calibration and tuning because it can be run with AGCMs using prescribed SSTs (Golaz et al., 2013; Zhao M. et al., 2018), and it is highly correlated with

the transient climate response and the equilibrium climate sensitivity of the same models run with coupled interactive ocean (Medeiros et al., 2014).

## 2.2    Perturbations to aerosol forcing and atmospheric parameters

We focus on two aerosol species, sulfate and black carbon (BC), and we perturb the radiative forcing from both species simultaneously. Current best estimates of aerosol radiative forcing (ARF) are -1.9 to 0.1 W m$^{-2}$ with medium confidence

(Boucher et al., 2013), with by far the dominant source of uncertainty arising from interactions with clouds (Stevens, 2015; Regayre et al., 2018). Sulfate exerts a well-known and strong negative radiative forcing on climate due to direct scattering of solar radiation, and through interactions with cloud properties (Boucher et al., 2013). By contrast, BC is an absorbing aerosol that generally exerts a positive ARF, although the sign depends on whether the BC layers are above (negative) or below (positive) cloud layers (Kim et al., 2015; Regayre et al., 2018; Ban-Weiss et al., 2011). The largest uncertainties in BC ARF are

due to an incomplete inventory of emissions, as well as poor understanding of aging/scavenging rates, vertical and horizontal transport, and deposition (e.g., Bond et al., 2013).

To examine uncertainty due to sulfate and BC forcing in CAM4, we introduced four new parameters to the model ($x_1$–$x_4$, see Table 1), with $x_1$ controlling sulfate and $x_2$–$x_4$ controlling BC. CAM4 does not include aerosol-cloud interactions, yet sulfate aerosols are known to be effective cloud condensation nuclei. As a proxy, parameter $x_1$ attempts to mimic these

interactions by specifying the fraction of the sulfate mass that uses optical properties for sulfate in "hygroscopic mode," i.e., a pure $SO_4$ molecule grown hygroscopically (following Köhler theory) corresponding to a relative humidity of 99%. The optical properties for the resulting sulfate aerosol (single scattering albedo, asymmetry parameter, and extinction coefficient) are given by $k_{\mathrm{default}}(1-x_1)+k_{\mathrm{hygro}}x_1$, where $k_{\mathrm{default}}$ indicates the default sulfate aerosol parameters in CAM4, and $k_{\mathrm{hygro}}$ indicates the parameters corresponding to this hygroscopic aerosol. This procedure ensures that the total mass of sulfate, and its geographic

location, are the same in all cases, and the only perturbation is the fraction of sulfate that is water-coated (regardless of the atmospheric humidity profile in the vicinity of the sulfate molecules).

To perturb BC forcing we assume that the optical properties are known, but that the total mass and horizontal and vertical distributions are uncertain, broadly consistent with the conclusions of Bond et al. (2013). We define three parameters $x_2$ (horizontal distribution), $x_3$ (mass scaling) and $x_4$ (vertical distribution) that control the distribution and amount of BC mass in

the model. More specifically, $x_2$ describes linear interpolation between the default horizontal distribution for BC mass ($x_2 = 0$) and BC being uniformly distributed throughout the globe ($x_2 = 1$). This parameter addresses the uncertainty associated with BC aging and transport: the longer that BC particles are able to survive in the atmosphere without being scavenged, the further the distribution should spread into pristine marine and polar environments. $x_3$ takes values between 0 and 40 and simply serves as a multiplier on the BC distribution, indicating uncertainties in total emissions and hence total mass loading. $x_4$ corresponds





to an altitude (0–40 km), indicating where a "layer" of BC (with mass equal to the total default mass) is added to the model, and then the total mass is rescaled to be the appropriate value per parameter $x_3$. This parameter addresses uncertainties associated with large-scale transport of BC by the atmospheric circulation, which is known to be poorly simulated (Lamarque et al., 2010).

We also investigate the sensitivity to five uncertain atmospheric parameters in CESM1-CAM4 that are related to clouds and convection. The parameters (denoted $x_5$–$x_9$) were identified as highly important in a previous one-at-a-time sensitivity analysis (Covey et al., 2013), and are described in Table 1. Two parameters ($x_5$ and $x_8$) control the threshold of atmospheric relative humidity that must be achieved before low and high clouds form, respectively; increasing either of these parameters will reduce the amount of low, or high, cloud in the model. Parameter $x_6$ changes the radius of liquid cloud droplets over the ocean, with smaller radii associated with brighter marine clouds, that are known to be highly important for climate sensitivity (Stevens, 2015; Sherwood et al., 2014). Parameters $x_7$ and $x_9$ are the timescales for shallow and deep convection, respectively; increasing either parameter will result in longer-duration convective precipitation. These parameters exert a large control on the mean climate in CAM4, but they are also expected to influence the climate sensitivity (Gent et al., 2011; Bony et al., 2015; Sherwood et al., 2014). We therefore vary these five atmospheric parameters in tandem with changes to the aerosol forcing to identify plausible climates with different climate sensitivities.

## 2.3 Emulation

For the nine parameters described in Section 2.2, we would need to perform at least O[$10^5$] simulations with CAM4 to adequately sample the parameter space in a typical all-at-a-time mode. Even running CAM4 at relatively low resolution, this would be impractical. The solution is to train an efficient statistical emulator of the dynamical model, which can be used to predict the climate output for any combination of parameter values, provided that the parameters lie within the range over which the emulator has been trained.

Following Lee et al. (2012) and McNeall et al. (2016), we construct a Gaussian Process (GP) emulator of CAM4 using the R package diceKriging (Roustant et al., 2012), which fits an $N$-dimensional nonlinear regression model to predict an output $y$ based on a series of $k$ predictors (input parameters $x_1$–$x_9$). Alternative methods of emulation have been used for climate modeling applications, including generalized linear models (e.g., Yang et al., 2017) and artificial neural networks (Sanderson et al., 2008). However, the GP model has two attractive properties that make it highly applicable to this type of problem: it can capture nonlinear interactions between the output and multiple inputs, and it provides an estimate of its posterior uncertainty.

We begin by defining $n = 350$ combinations of parameter values ($x_1$–$x_9$) for the training points in the 9-parameter space, using a Latin Hypercube design that ensures good distribution of cases, even in the corners of the hypercube (McKay et al., 1979). For each training point, we produce three one-year realizations of CAM4, each using different atmospheric and oceanic initial conditions drawn from a 500-year control integration of the coupled ocean-atmosphere version of CAM4 at the same resolution (see Section 2.1). The mean of the resulting outputs from the three realizations is used to train the emulator, which reduces the noise arising from internal climate variability. To quantify the impact of the parameters on climate sensitivity, the training process must be repeated twice: once using prescribed preindustrial SSTs, and then again using warmed SSTs (see Section 2.1). For each training point, the necessary simulations take approximately six hours on a single 8-core node of a high



**Table 1.** List of parameters that are perturbed in this study, including for each parameter a description, the range of perturbed values, and the default value in CAM4 (where applicable).

| parameter | description (CAM4 parameter name) | min | default | max | notes |
|---|---|---|---|---|---|
| $x_1$ | Fraction of hygroscopic $SO_4$ | 0.0 | 0.0 | 1.0 | Proxy for sulfate indirect effect (no units). |
| $x_2$ | Spatial uniformity of BC (1 = globally uniform) | 0.0 | 0.0 | 1.0 | Proxy for BC aging and scavenging (no units). |
| $x_3$ | Scaling factor for global BC mass | 0.0 | 1.0 | 40.0 | Proxy for uncertainty in BC emissions (no units) |
| $x_4$ | Altitude for insertion of uniform BC layer | 0.0 | – | 39.0 | Proxy for vertical transport of BC (units km). Note: new parameter, no default. |
| $x_5$ | RH threshold for low cloud formation (cldfrc_rhminl) | 0.80 | 0.88 | 0.99 | Value grid box RH must exceed before low cloud forms (no units) |
| $x_6$ | Effective radius of liquid cloud droplets over ocean (cldopt_rliqocean) | 8.4 | 14.0 | 19.6 | (units microns) |
| $x_7$ | Timescale for consumption rate of shallow CAPE (hkconv_cmftau) | 900 | 1800 | 14440 | (units seconds) |
| $x_8$ | RH threshold for high cloud formation (cldfrc_rhminh) | 0.50 | 0.50 | 0.85 | Value grid box RH must exceed before high cloud forms (no units) |
| $x_9$ | Timescale for consumption rate of deep CAPE (zmconv_tau) | 1800 | 3600 | 28800 | (units seconds) |

performance computing cluster, giving a total computing time of $350 \times 6 \times 8 = 17,000$ core hours. Applying the emulator to predict an output variable at $10^5$ or $10^6$ uniformly sampled points in parameter space takes less than 30 seconds on a single core of a modern desktop computer. This is many orders of magnitude faster than it would take to run a set of $10^5$ or $10^6$ simulations with the dynamical model.

5    The output from the training simulations are used to construct the emulator. We make standard assumptions to configure the GP model, using a linear prior and the default Matérn covariance function (Roustant et al., 2012; Lee et al., 2011), although we have verified that our conclusions are insensitive to these choices. Before the emulator can be used for prediction, its performance is validated using leave-one-out cross-validation (LOOCV): each case from the $n = 350$ training set is left out in turn, and the emulator is rebuilt using the remaining $n = 349$ cases. The resulting model is then used to predict the output for

10   the case that was left out, and so on until each case has been predicted. The performance metric used here is the correlation coefficient between the $n = 350$ outputs simulated by CAM4 and the $n = 350$ predictions from the emulator run in LOOCV mode. The validation results are presented in Section 4.1.



## 2.4 Quantifying the plausibility of candidate models

The plausibility of the climate produced by a particular combination of input parameters is assessed using a multivariate skill score (SS), based on Pierce et al. (2009):

$$SS_X = r_{p,d}^2 - \left[ r_{p,d} - (\sigma_p/\sigma_d) \right]^2 - \left[ (\bar{p} - \bar{d})/\sigma_d \right]^2 \tag{1}$$

where for a spatial grid of particular output variable $X$ (e.g., precipitation, low cloud amount, etc.), $p$ denotes a perturbed model, and $d$ denotes the default (reference) model, $r_{p,d}$ is the anomaly (pattern) correlation between $X$ in $p$ and $d$, $\sigma$ is the spatial standard deviation of $X$, and overbars denote the spatial mean of $X$.

The SS quantifies the mean bias, spatial correlation, and spatial variance of six key simulated variables for each perturbed model relative to the default version of CAM4. The variables included in SS are low cloud fraction (CLDL), total precipitation (PRECT), net TOA radiative flux (FNET), shortwave cloud forcing (SWCF), longwave cloud forcing (LWCF), and global vertically-integrated longwave heating rate (QRL). We calculate SS for each variable separately, and then average the SS values to obtain the final SS for each perturbed model. To obtain a high value of SS (SS∼1), a parameter combination must produce a simulated climate that is simultaneously close to that of the default model in all of these fields. We apply a stringent threshold of SS > 0.85 to determine whether a particular perturbed model is plausible, which equates to approximately the 85th percentile of the SS distribution. We compute SS for the n=350 training cases, and then use the emulator to predict SS for all possible parameter combinations (see Section 4.1).

## 3 Controls on climate sensitivity in the CAM4 training simulations

### 3.1 Relationship between inputs and outputs

Figure 1 presents the relationships between all inputs (perturbed parameters $x_1$–$x_9$) and all outputs, for the $n = 350$ training simulations run with CAM4. The Latin Hypercube sampling of input parameters (Section 2.3) ensures an even sampling of values across each input parameter's full range (see Table 1). Correlations between parameter values are very weak, which provides confidence that each training case is an independent event drawn from the parameter population. The default values in CAM4 of parameters $x_5$ and $x_6$ are located within the center of their distributions, while the values for aerosol parameters $x_1$–$x_3$, and atmospheric parameters $x_7$–$x_9$, are at the lower end. For the sensitivity parameter for high clouds ($x_8$), the default is also the minimum value (0.5); however, we note that the default value varies considerably with horizontal resolution, and at the 2-degree resolution used, for example, by Covey et al. (2013), its value is 0.80. No default exists in CAM4 for the new BC altitude parameter $x_4$.

The marginal relationships between inputs and outputs can provide an indication of which parameters may be important for emulating the outputs. Intuitively, the aerosol optical depth output (AOD) is a strong function of the sulfate hygroscopic fraction ($x_1$) and BC mass scaling ($x_3$), but shows no obvious relationships with the other aerosol-related parameters ($x_2$ or $x_4$), and most other atmospheric outputs are not strongly correlated with any of the aerosol parameters. One exception is



total precipitation, which appears to decrease as a function of BC mass scaling ($x_3$), presumably because of a reduction in precipitating clouds induced by so-called semi-direct effects (Bond et al., 2013). For the atmospheric parameters, there are clear relationships between inputs and the output variable(s) directly related to the parameter being perturbed. For example, there is a strong negative relationship between low cloud amount (CLDL) and the sensitivity parameter for low clouds ($x_5$), and

longwave cloud forcing (LWCF) is negatively correlated with the sensitivity parameter for high clouds ($x_8$). Most apparent from Fig. 1 is the interconnectedness of the outputs; for example, low cloud, net radiation and shortwave cloud forcing are highly correlated. Parameter $x_5$ strongly affects CLDL, and this produces knock-on effects to all other outputs: positive correlations with FNET, QRL and SWCF, and negative correlations with LWCF and PRECT.

The multivariate skill score (SS, see Section 2.4) shows no obvious relationships with any aerosol parameters, nor with the

majority of the atmospheric parameters. Exceptions are the nonlinear relationship with $x_5$ (higher values of $x_5$ are inconsistent with low SS), and the deep convective timescale ($x_9$), which is negatively correlated with, and sets an upper bound on, SS. This suggests that it is not feasible to achieve a climate that is consistent with the default model when the timescale for deep convection is longer than 3–4 hours (the default value of $x_9$ in CAM4 is 1 hour). Similar results are seen for the Cess climate sensitivity (CS), which shows no relationship to the aerosol parameters or atmospheric parameters $x_6$–$x_8$. A positive correlation

is found between CS and $x_5$: higher $x_5$ produces less low cloud, and high-$x_5$ cases are, on average, slightly more sensitive to warming, which is consistent with expectations based on comprehensive ESMs (e.g., Siler et al., 2018). Lastly, we note an interesting nonlinear impact of $x_9$ on CS, where CS increases linearly with $x_9$ up to about a 4-hour timescale, then becomes much more variable and begins to decrease again.

### 3.2 Probability distribution of output variables

The black lines in Fig. 2 show the distribution of global mean outputs based on the $n = 350$ training cases from CAM4, for the six variables that comprise the SS, and the Cess CS. The six variables are expressed as biases relative to the default model, so that a value of zero represents a case that perfectly reproduces the default model's global mean for that variable. Their distributions are unimodal and straddle zero, indicating that the majority of cases produce climates reasonably close to the default CAM4. However, for most variables the tails of the distribution are much longer than a Gaussian, indicating a

large number of climates that are far from the default model. Indeed, the full range of climates produced in this ensemble is dramatic: the spread in top of atmosphere (TOA) net radiative flux (FNET) is between $-30$ and $+20$ W m$^{-2}$, which is an order of magnitude larger than the FNET response expected due to a doubling of the atmospheric $CO_2$ concentration (Andrews et al., 2012). The distributions of CLDL and FNET are also shifted toward positive and negative values, respectively, indicating a tendency for the perturbations to increase low cloud in the model, increasing shortwave flux to space. Fig. 1 shows that

this effect is controlled almost entirely by parameter $x_5$, the sensitivity parameter for low clouds, which has previously been identified as very important in CAM4 (Covey et al., 2013). The impact of $x_5$ may also be asymmetric: reducing $x_5$ by 0.01 from its default value of 0.88 is likely to have a greater impact on cloudiness than increasing $x_5$ by the same amount, because $x_5$ is a relative humidity (RH) threshold, and the distribution of RH is heavily skewed toward lower values.





**Figure 1.** Scatterplot matrix showing the relationship between all input parameters ($x_1$–$x_9$), and all output variables (names defined in the text) in the $n = 350$ training cases run with CAM4. Red points show the default values for all input parameters except $x_4$, which has no default.





The distribution of SS is bounded by 0 and 1, by construction, with a single peak at around 0.8, a maximum SS of 0.953, and a long left tail. The peak in the distribution of SS at 0.8, and the fact that $\max(\mathrm{SS}) < 1.0$, implies that all the cases within our ensemble are—to a greater or lesser extent—imperfect representations of the default model. This is mostly explained by the parameter $x_4$ (the altitude of injection of a uniform layer of BC): no perturbed case can produce a climate exactly like

the default model, because the default model does not include $x_4$, and the experimental design specifies that a BC layer is always injected *somewhere* between 0–40 km. We reiterate that the emphasis here is on identifying plausible candidate models with altered aerosol and atmospheric parameters, not on tuning/calibration to make the default model more *realistic* (relative to observations). The distribution of CS shows a range between $0.35 - 0.65$ K/Wm$^{-2}$, with the default value of 0.45 K/Wm$^{-2}$ located at around the 10th percentile. Therefore, around 90 % of all candidate models produce a higher climate sensitivity than

the default CAM4. The input parameters driving these changes are explored in Section 5.

## 4   Exploring the parameter space through emulation

### 4.1   Validating the emulator

The sample of $n = 350$ training cases provides a large ensemble of cases (relative to modern ensembles with comprehensive ESMs Kay et al. (2014)) with which to study the effects on CS from aerosol forcing and atmospheric parameters (Figs. 1–2).

However, the nine parameters $x_1$–$x_9$ map onto a vast parameter space that is computationally impractical to sample adequately using CAM4 itself. A more practical way to explore the response space (i.e., to fill in the unsampled regions for the output variables in Fig. 1) is by using a statistical emulator.

Using the emulator of CAM4 described in Section 2.3, we make predictions for each output variable shown in Fig. 2 using fine-resolution uniform sampling over the full range of each parameter $x_1$–$x_9$. The emulated results are shown as the gray

shaded regions in Fig. 2, and it is immediately apparent that the emulator does a very good job at reproducing the simulated distribution for each variable. The close agreement in all cases indicates that the uncertainty in the emulator is small, which provides confidence that the emulator is a useful tool to explore the parameter space of CAM4. However, we first conduct a more quantitative validation of the emulator, by performing leave-one-out cross-validation (LOOCV) to sequentially leave out, and then predict, each individual case from the $n = 350$ training sample. The results of this LOOCV procedure are shown

in Fig. 3, and reveal that the emulator is, in general, highly successful at predicting model outputs for unseen parameter combinations.

Taking into account emulator uncertainty (shown by the 95% confidence interval on each prediction), the agreement between the emulated and simulated values of the multivariate skill score (SS) is close to perfect ($r > 0.98$). For the Cess climate sensitivity (CS), while there is greater uncertainty on each prediction (margin of error $\sim$0.05 K/Wm$^{-2}$), and the emulated

values show a general tendency to be slightly underpredicted, RMSE is very low (0.023 K/Wm$^{-2}$) and the correlation skill is high ($r = 0.86$). A final demonstration of the value of using the Gaussian Process emulator is shown by repeating the emulation using a multiple least-squares linear regression (MLR) model. The righthand panels of Fig. 3 show that the MLR emulator performs substantially worse than the GP emulator in terms of both RMSE and correlation skill.



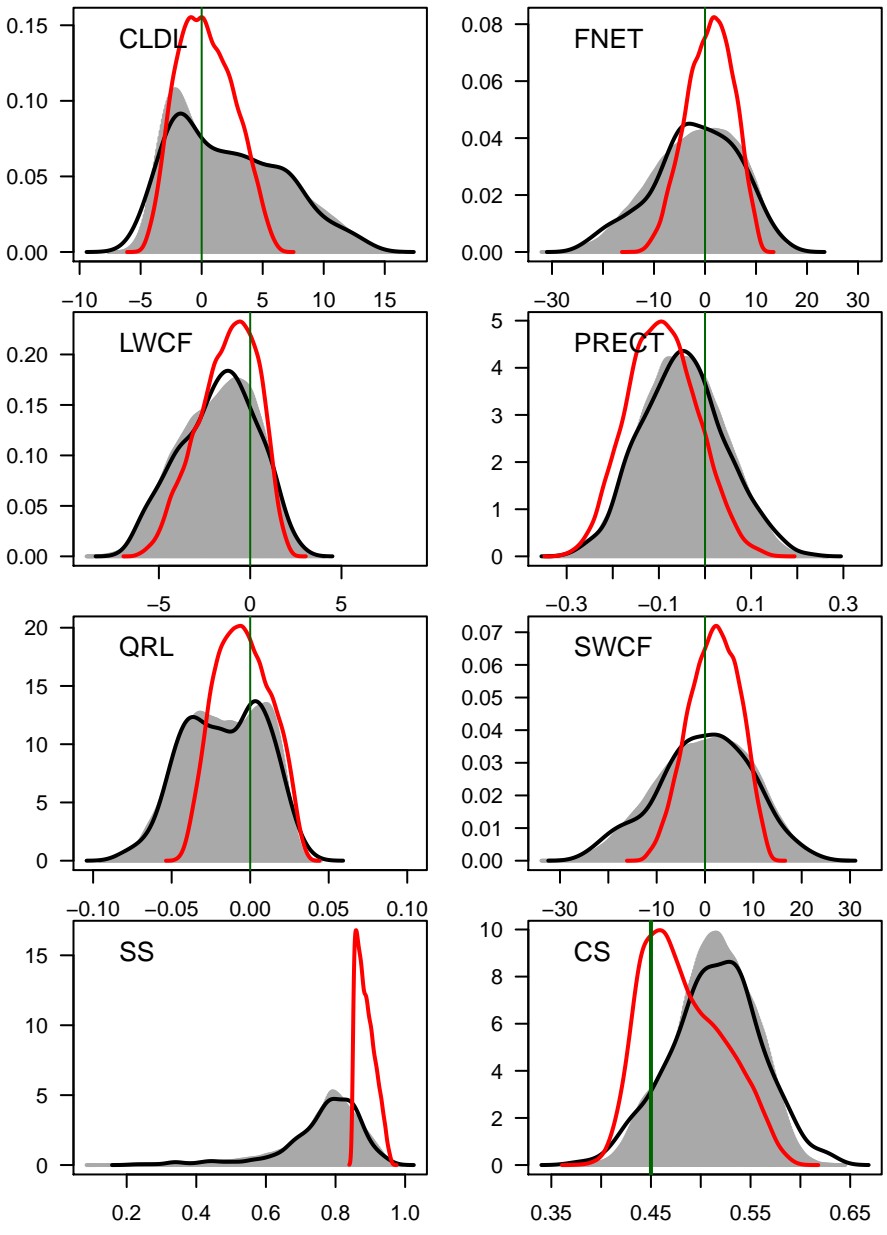

**Figure 2.** The black line shows the density distribution of simulated values from CAM4 for each output variable in the $n = 350$ training cases (variable names and units are defined in the text), the multivariate skill score (SS; unitless) and Cess climate sensitivity (CS; units K/Wm$^{-2}$). The gray shaded region shows the density distribution from 100,000 emulated samples of each output variable, and the red line shows the density distribution for the subset of emulated samples that is most plausible (see Section 4.3). Note the different x-axis scales in each panel. The dark green vertical lines indicate the location of the default model in all panels except SS.



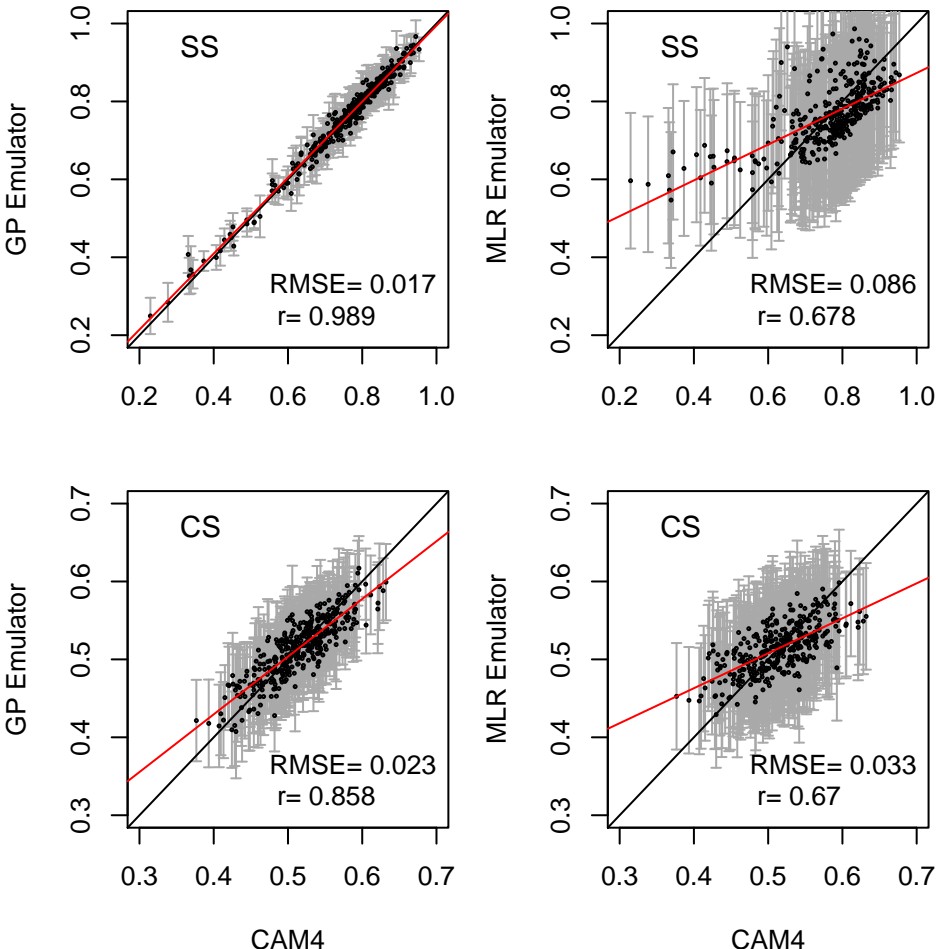

**Figure 3.** Points show predictions from the emulator ($y$-axis) against CAM4 simulated values ($x$-axis) for SS (top) and CS (bottom) for each case in the $n = 350$ training set, using the statistical emulator in LOOCV mode (see Section 4.1). Gray error bars show the 95% prediction interval for each case. The red line is a least-squares linear fit of the predicted against simulated values, and the black solid line shows the 1:1 line. The lefthand panels show predictions made using the GP emulator, and the righthand panels show predictions made using the MLR emulator.



## 4.2 Parameter sensitivity / importance

Parameter sensitivity is quantified following Carslaw et al. (2013) and McNeall et al. (2016) using the so-called FAST methodology (Saltelli et al., 1999), in the R package *sensitivity* (Pujol et al., 2017). This method separates the contribution to the total response from "main effects", that are directly attributable to variations in each (normalized) parameter, and interactions between parameters, which are calculated as the residual: interaction = total − main effect.

Figure 4 reveals that atmospheric parameters $x_5$ and $x_9$ are most influential for the outputs SS and CS, explaining a combined total of ∼75% of the total variance in each output. The variation in output AOD is explained almost entirely by $x_1$ (sulfate hygroscopic fraction), with a small residual contribution by $x_3$ (BC mass scaling). No other aerosol parameters are influential for any of the output variables shown, and there are small (< 10%) contributions from atmospheric parameters $x_6$−$x_8$ for SS and CS. In general, the main effects are dominant; however, non-negligible parameter interactions are found for SS, where they make up almost half of the total variance explained by $x_5$. While difficult to directly interpret, we hypothesize that this emphasis on the interaction terms is due to the interrelated nature of the parameters $x_5$−$x_9$, all of which influence clouds, precipitation, and radiative flux (i.e., all of the variables assessed in computing SS) in some form. These results agree closely with previous work examining the sensitivity to atmospheric parameters in CAM4. For example, Covey et al. (2013) show that $x_5$ and $x_9$ are highly influential parameters for top-of-atmosphere radiative flux, although their one-at-a-time methodology did not permit an examination of parameter interactions.

## 4.3 Identifying a plausible set of input parameters

In this section, we use the statistical emulator to identify regions of the 9-dimensional parameter space that produce plausible climates, which are defined as those similar to the climate of the default CAM4. We begin by using the emulator to construct a set of $n = 100{,}000$ cases for output variables SS, AOD and Cess CS, based on a uniform sample of the distributions of each parameter ($x_1$−$x_9$). Applying first the threshold SS>0.85 eliminates ∼85% of cases, and adding a second constraint (AOD<0.08) eliminates a further 6% of cases whose AOD is too far from that of default CAM4. The threshold for AOD represents a trade-off between finding a sufficiently large sample of cases, and their fidelity to the default model. Since present-day AOD in default CAM4 tends to be biased low [satellite observations from MODIS+MISR for present day show AOD∼0.16, compared to 0.11 for CAM4; Remer et al. (2008)], and the aerosol perturbations $x_1$−$x_4$ tend to increase AOD, the threshold ensures that plausible cases maintain a global mean AOD that is within 50% uncertainty of the default CAM4. After applying both thresholds, only ∼9% of the original parameter space remains plausible, and the density distributions of parameters for this remaining space are shown in Fig. 5.

Next we attempt to constrain the parameter ranges by examining the regions of parameter space in Fig. 5 that produce higher/lower densities of plausible outputs. The only aerosol parameter that can be constrained is $x_1$, where all values above 0.6 are implausible. The BC mass scaling parameter $x_3$ shows a slightly reduced density of plausible cases for very high BC mass; however, even very high BC mass cannot be ruled out completely, because 15% of cases remain plausible with $x_3 > 32$. For the atmospheric parameters ($x_5$−$x_9$), the range of $x_5$ is compressed toward a central value that is not far from, but with





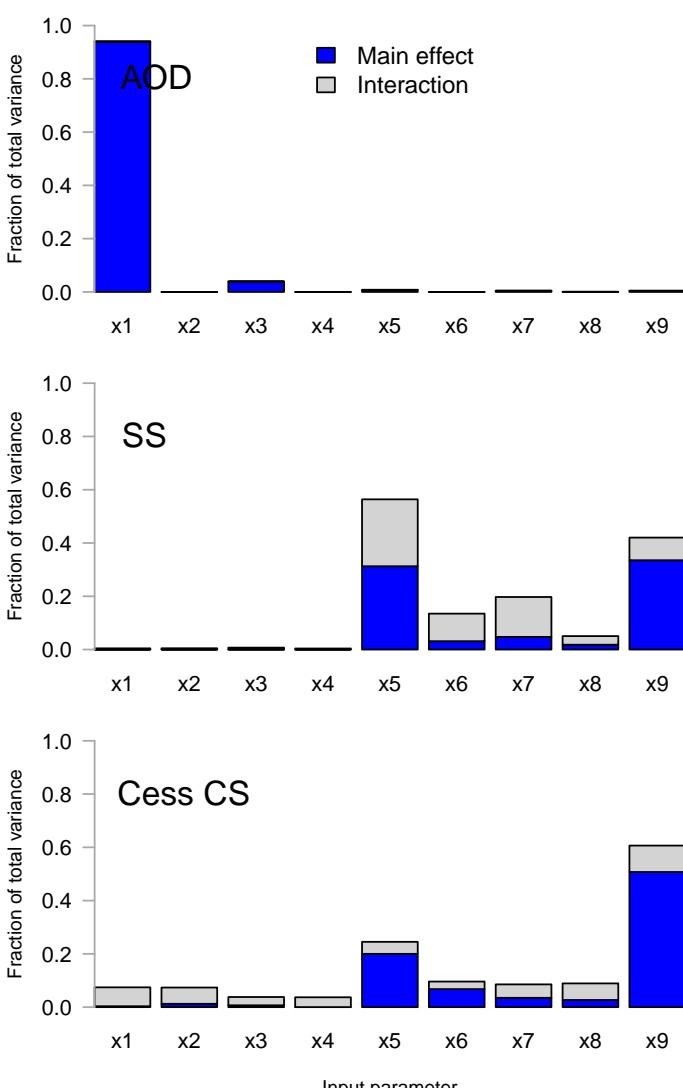

**Figure 4.** Fraction of total variance explained by the each input parameter, for AOD (top), SS (middle) and Cess CS (bottom). The light gray portion of each bar represents the contribution of interaction terms, while the blue portion represents the main effect. The full height of each bar represents the total variance explained by each input parameter, with more influential parameters explaining a higher fraction of variance.





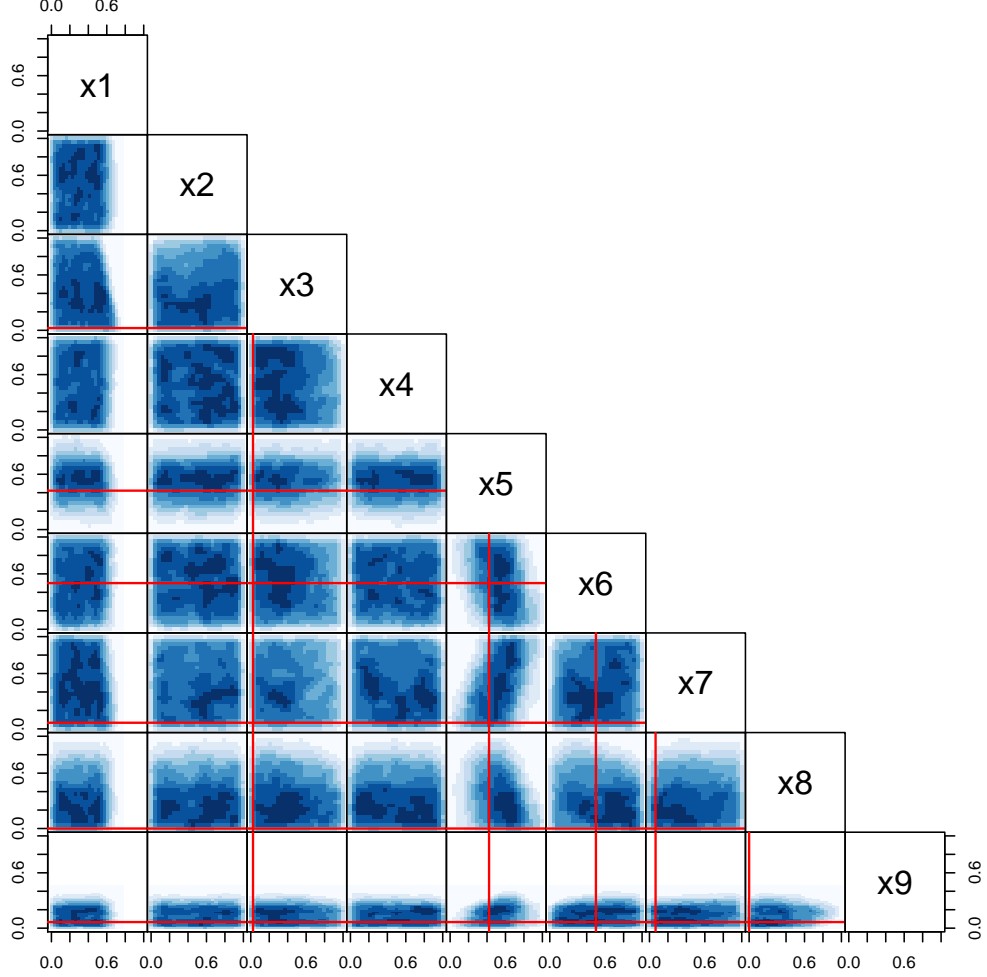

**Figure 5.** Pairs density plots for input parameters, showing the density of plausible cases, defined as the subset of $n = 100,000$ emulated samples that have SS>0.85, and AOD<0.08. The default values for each parameter in CAM4 are indicated by red lines in the panels.

a tendency to be higher than, the default value in CAM4 (denoted by the red lines in Fig. 5). For $x_8$ (high cloud sensitivity) there is a tendency for most plausible cases to have higher parameter values than default, but very high values are ruled out. The parameter $x_9$ (deep CAPE timescale) is dramatically constrained from its original range: all values above ∼4 hours are ruled out, and the default value is located well within the plausible range. Parameters $x_6$ (liquid drop radius over ocean) and

5    $x_7$ (shallow CAPE timescale) do not show any obvious reduction in their plausible ranges.

Finally, we examine the emulated outputs associated with the subset of plausible cases. The red line in each panel of Fig. 2 shows the distribution of output values only for the plausible cases, which provides an estimate of how much our candidate



model versions differ from the default CAM4. For all variables the spread for the plausible subset is often considerably smaller than the spread for all cases, and tends to be shifted toward a mean of zero (a perfect representation of the global mean from the default model). This indicates that our threshold-based approach to plausibility is working as desired: the outputs from the plausible models *should* be closer to the default model, by construction. Interestingly, high values of CS become much less

likely after imposing the thresholds. It could be argued that CS is directly influenced by the variables in the SS, and so it is not independent of our thresholding approach; however, we consider CS to be an emergent property of the model and, therefore, this result could not have been predicted a priori.

## 5   Impact on climate sensitivity

Running the CAM4 model with its default (unperturbed) settings for parameters $x_5$–$x_9$, and without perturbations to the

additional aerosol parameters $x_1$–$x_4$, we find a Cess CS value of 0.45 K/Wm$^{-2}$. The median value of CS for all $n = 100,000$ emulated cases is 0.51 K/Wm$^{-2}$, which implies that the net effect of the perturbations to the aerosol parameters ($x_1$–$x_4$) is to increase CS. The 95% interval of CS values for the 9% of emulated cases that are plausible is 0.418 K/W m$^{-2}$ (7% lower than default) to 0.538 K/W m$^{-2}$ (20% higher than default). This shows that, to some extent, CS is a tunable quantity; however, this range is only about 25% of the range in CS across an ensemble of CMIP5 models by Medeiros et al. (2014), suggesting that

structural uncertainty (not sampled here) also plays a major role in contributing to the spread of simulated CS.

We next examine the distribution of aerosol and atmospheric parameters that are associated with high (CS $> 0.538$ K/Wm$^{-2}$) and low (CS $< 0.418$ K/Wm$^{-2}$) sensitivity cases; i.e., cases with CS in the upper and lower 2.5% of all plausible cases. Figures 7-8 show little difference in the distribution of the aerosol parameters ($x_1$–$x_4$) for high or low sensitivity, suggesting that neither the hygroscopicity of sulfate, nor the mass and spatial distribution of BC, are important for determining CS in

this model. Much larger differences between the high and low sensitivity cases are found for the atmospheric parameters. As suggested from Fig. 1, high sensitivity is associated with higher values of $x_5$ (less low cloud), $x_8$ (less high cloud) and $x_9$ (longer deep convection), and lower values of $x_6$ (smaller liquid cloud droplets). The parameter $x_7$ (shallow CAPE timescale) appears to have little influence on CS. The situation for low sensitivity is broadly the inverse, and the narrow ranges for some parameters (e.g., $x_5$ and $x_9$) provide clear constraints on radiative-convective processes that control climate sensitivity in this

model. The speckled blue-yellow-red nature of the panels for $x_1$–$x_4$ in Fig. 8 shows that the spread of CS is very similar for all values of the aerosol parameters. This suggests that, in tandem with the right combination of atmospheric parameters, any value of CS within the model's range can be achieved for any strength of aerosol forcing. Figure 8 also reveals multiple effects of $x_9$ on CS: the main effect (c.f. Fig. 4) is a strong linear gradient in CS from low to high parameter values, in addition to clear nonlinear interactions with parameters $x_6$ and $x_8$.

Collectively, these results suggest that our overall objective of configuring different, but equally plausible, versions of CAM4 with varying strengths of aerosol forcing and Cess CS, is eminently achievable. To this end, we conclude this Section by presenting examples in Table 2 of parameters with very different aerosol forcings selected from the emulated cases, which produce CS values across the full range for this model. To extract these cases we apply joint thresholds to parameters $x_1$





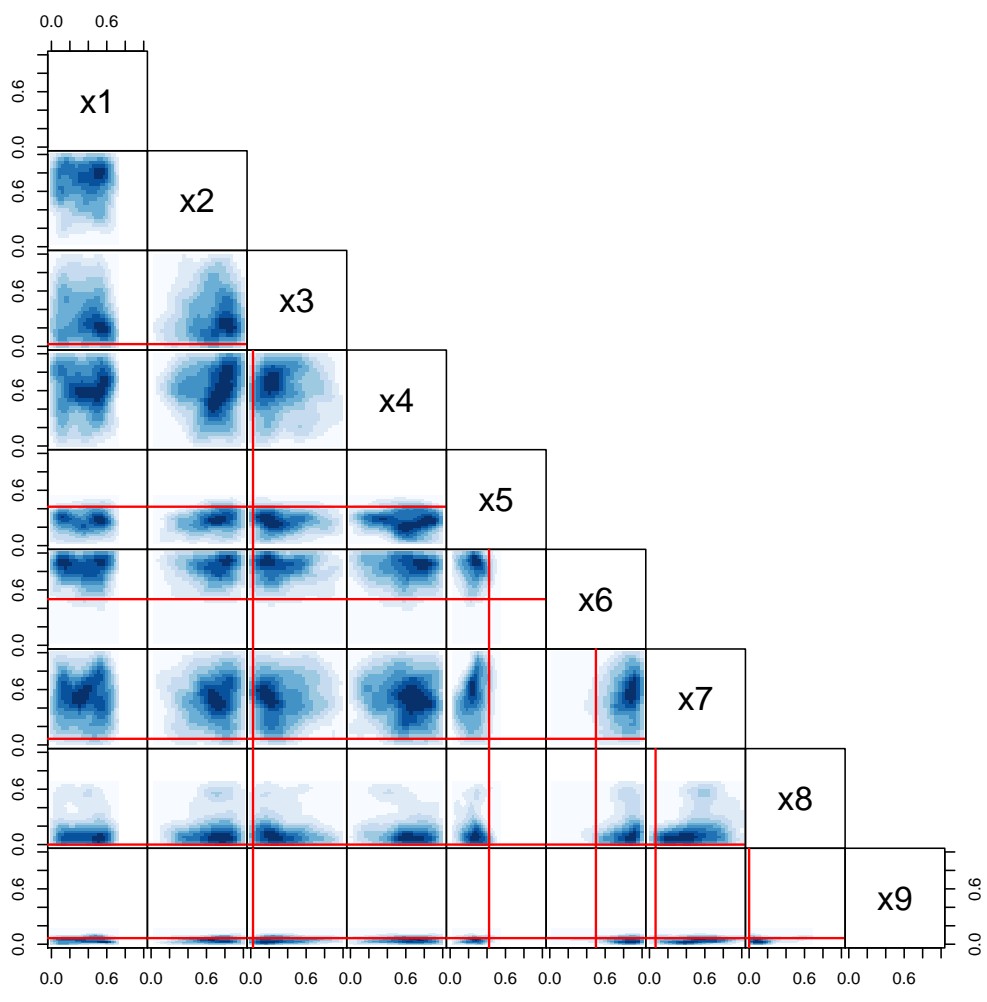

**Figure 6.** As Fig. 5, except only for the cases in the lowest 2.5% of the distribution of Cess climate sensitivity ($<0.418$ K/Wm$^{-2}$).





**Table 2.** Each row shows the input parameter values (columns 2-10), and the Cess climate sensitivity (CS, units K/Wm$^{-2}$; column 11) predicted by the emulator for that combination of parameters. The first column denotes the strength of aerosol radiative forcing (ARF) in a case from sulfate (S) and Black Carbon (B): h=high, m=medium and l=low (defined in the text). For reference, the default parameter values, and associated CS, are shown near the top.

|  | $x_1$ | $x_2$ | $x_3$ | $x_4$ | $x_5$ | $x_6$ | $x_7$ | $x_8$ | $x_9$ | CS |
|---|---|---|---|---|---|---|---|---|---|---|
| *Default settings* | | | | | | | | | | |
|  | 0.0 | 0.0 | 1.0 | – | 0.88 | 14.0 | 1800 | 0.50 | 3600 | 0.45 |
| *High Sensitivity Cases* | | | | | | | | | | |
| Sh.Bl | 0.50 | 0.05 | 0.9 | 37.9 | 0.98 | 9.3 | 9870 | 0.63 | 13066 | 0.61 |
| Sh.Bh | 0.52 | 0.48 | 23.7 | 28.3 | 0.93 | 10.5 | 1860 | 0.63 | 13202 | 0.58 |
| Sm.Bm | 0.20 | 0.08 | 11.7 | 38.0 | 0.97 | 9.1 | 12279 | 0.65 | 12759 | 0.60 |
| Sl.Bl | 0.02 | 0.24 | 3.9 | 0.7 | 0.95 | 11.3 | 4547 | 0.60 | 13753 | 0.57 |
| Sl.Bh | 0.04 | 0.62 | 20.7 | 4.2 | 0.99 | 10.3 | 7016 | 0.55 | 12297 | 0.59 |
| *Low Sensitivity Cases* | | | | | | | | | | |
| Sh.Bl | 0.55 | 0.47 | 3.1 | 21.0 | 0.82 | 14.7 | 4570 | 0.52 | 2524 | 0.39 |
| Sh.Bh | 0.58 | 0.74 | 26.2 | 31.4 | 0.84 | 18.0 | 8667 | 0.51 | 3741 | 0.40 |
| Sm.Bm | 0.22 | 0.89 | 10.2 | 4.1 | 0.90 | 18.2 | 12548 | 0.70 | 1896 | 0.40 |
| Sl.Bl | 0.08 | 0.48 | 2.9 | 38.1 | 0.86 | 15.4 | 2635 | 0.51 | 1824 | 0.41 |
| Sl.Bh | 0.08 | 0.54 | 22.1 | 36.7 | 0.83 | 18.1 | 6066 | 0.51 | 2185 | 0.39 |

(sulfate hygroscopicity) and $x_3$ (BC mass scaling) to identify combinations with high ("h") and low ("l") sulfate ("S") and BC ("B") forcing; for example, Sh.Bl denotes cases with high sulfate, and low BC, forcing. After extracting a distribution of cases for each combination of sulfate and BC forcing, we record the parameters that produce the minimum (low sensitivity), and maximum (high sensitivity), value of CS.

5    As expected from Fig. 8 and the discussion above, we are able to identify both high and low sensitivity cases for all combinations of aerosol forcing. Comparing pairs of rows for the same aerosol forcing at high and low sensitivity reveals that, by construction, they tend to have similar aerosol parameters. However, clear differences emerge in the atmospheric parameters: the high sensitivity cases tend to have higher $x_5$, $x_8$ and $x_9$, and lower $x_6$. Only $x_7$ appears unrelated to CS, perhaps due to its influence on shallow cumulus clouds, which are more likely to be overlain by higher clouds and, therefore, have limited

10   influence on the top-of-atmosphere energy budget. This table provides a prototype for a future study to test a suite of cases in CAM4 with a fully interactive ocean, to determine the relationship between ARF, Cess CS and the transient climate response (e.g., Golaz et al., 2013; Zhao M. et al., 2018).





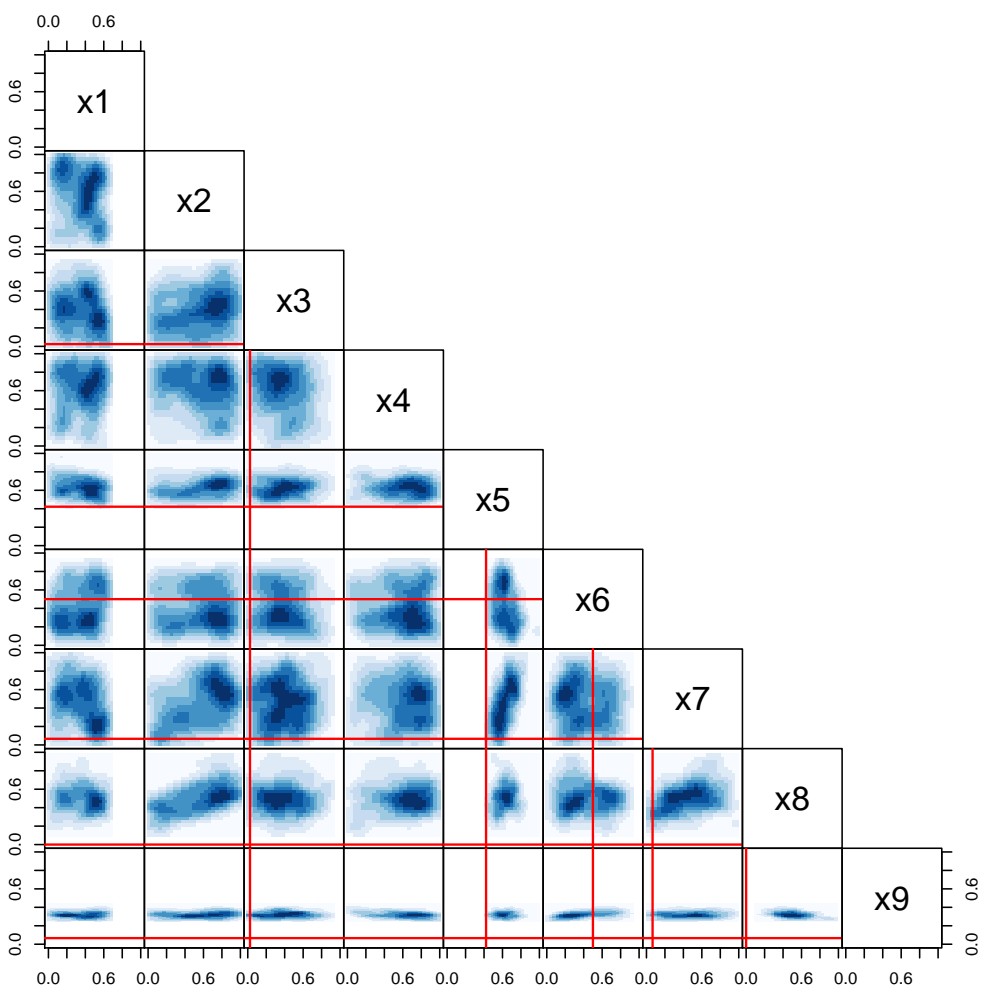

**Figure 7.** As Fig. 5, except only for the cases in the highest 2.5% of the distribution of Cess climate sensitivity ($>0.538$ K/Wm$^{-2}$).





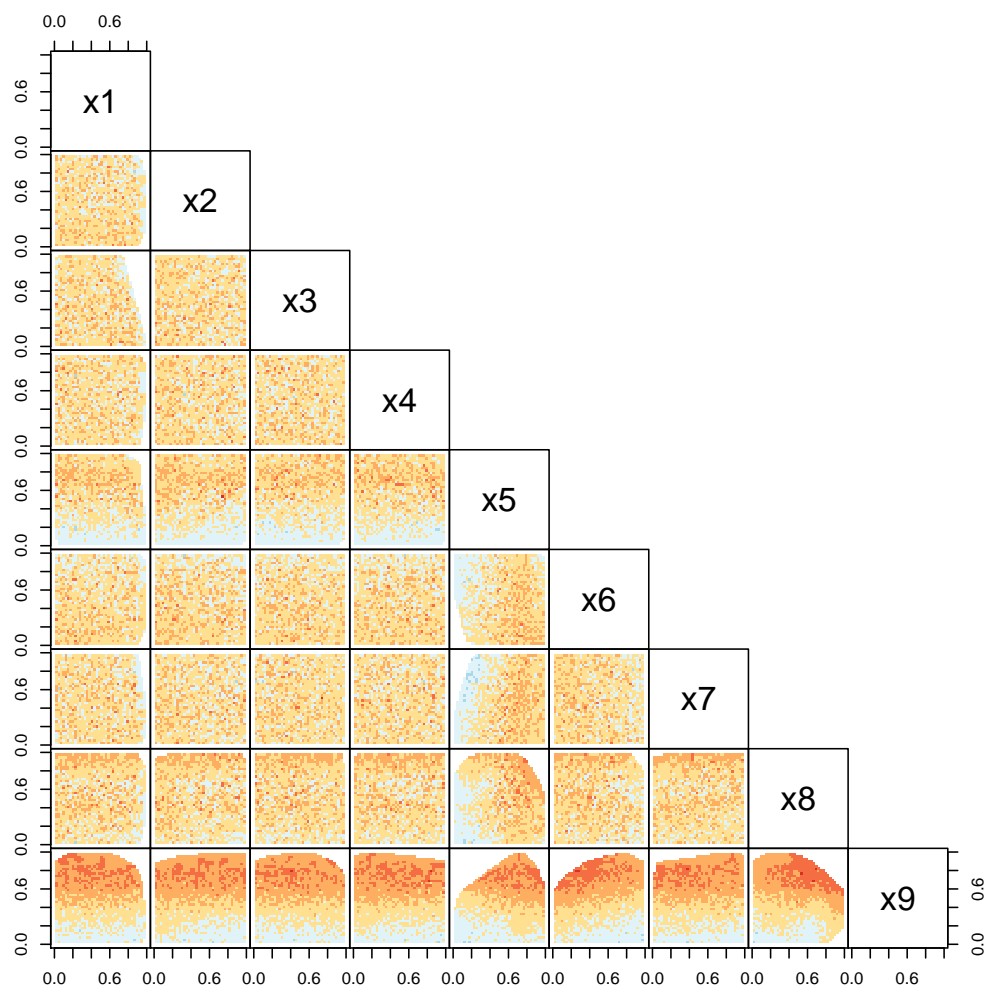

**Figure 8.** Color shading shows Cess climate sensitivity (K/Wm$^{-2}$) for plausible cases as a function of each pair of input parameters ($x_1-$ $x_9$). Red shades indicate higher climate sensitivity in the range 0.25-0.65 K/Wm$^{-2}$, and blue shades indicate lower climate sensitivity. White areas indicate regions of parameter space where there are no plausible cases.



## 6   Conclusions

We employ a statistical emulation procedure to sample the parameter/response space of the atmospheric general circulation model NCAR CESM-CAM4. The influence of four aerosol parameters controlling the aerosol radiative forcing (ARF) from sulfate and black carbon, and five atmospheric parameters controlling clouds and convection, are assessed in combination

across their full range of uncertainty. A multivariate skill score is used to determine the plausibility of each combination of parameters, and thus to constrain plausible parameter ranges, and the spread of an important emergent property of the model: its climate sensitivity (CS). We find that atmospheric parameters explain more than 85% of the variance in CS, and two parameters are most important: $x_5$ controls the amount of low cloud in the model, and $x_9$ controls the time scale for deep convection. The aerosol parameters have little impact on CS in our model configuration, making it equally possible to

identify cases with high/low ARF that have high, or low, CS (Table 2). However, while we attempt to quantify the impact of aerosol-cloud interactions (ACI) through the hygroscopicity parameter $x_1$, the CAM4 model does not include direct simulation of ACI, which could increase the importance of the aerosol parameters (Regayre et al., 2018). Future work should examine the importance of uncertainties in parameters related to subgrid-scale activation of cloud droplets in newer versions of the CESM-CAM models that include these processes, and quantify their impact on ACI and CS (Golaz et al., 2013).

Our results indicate that the climate sensitivity of CAM4 can be modified, and possibly constrained, through adjustments to select uncertain atmospheric parameters, primarily $x_5$ and $x_9$. These results can be compared with previous studies that examined the impact of tuning parameters on climate sensitivity (CS) in ESMs. We find a plausible spread of Cess CS values between 0.418 K/W m$^{-2}$ and 0.538 K/W m$^{-2}$, which spans approximately 25% of the range derived from a suite of CMIP5 models that performed a similar experiment (Medeiros et al., 2014). This is in good agreement with previous studies that

found that the spread in CS for a single ESM (either with interactive or prescribed ocean components) due to uncertain tuning parameters related to clouds and convection was smaller than the spread among the ensemble of CMIP models (Mauritsen et al., 2012; Golaz et al., 2013). This body of work, therefore, suggests that structural deficiencies in the configuration of ESMs contribute more to the uncertainty in CS than parametric uncertainty. Of the ∼25% of the spread in CS due to parametric uncertainty, our study indicates that atmospheric parameters explain the vast majority, with only a minor role for aerosol

parameters. The major new finding from this work is that a given model's position on the Kiehl curve *can* be varied through compensating adjustments to atmospheric parameters and radiative forcing, but only to a relatively small degree.

This study explores only the question of whether plausible alternative versions of CAM4 can be configured (through uncertain aerosol and atmospheric parameters) to have different climate sensitivities, relative to CAM4 at the same horizontal resolution with its default parameter settings. Using the default model to determine plausibility explicitly avoids the question

of plausibility relative to observations, or finding parameter combinations to "improve" CAM4. That would be an exercise in model tuning/calibration, which is beyond the scope of this study. However, our opinion is that the range of plausible solutions that have been revealed through the emulation procedure makes it highly likely that parameter combinations exist within the sampled parameter/response space that provide better matches to the observed climate than the default settings. This hypothesis will be examined in a future study.



*Data availability.* All data and scripts will be made available through the lead author's GitHub repository at: https://github.com/UWFletcherGroup.

*Competing interests.* The authors declare that they have no conflict of interest.

*Acknowledgements.* CF and BB were supported by the Network on Climate and Aerosols: Addressing Key Uncertainties in Remote Canadian Environments. The authors are grateful for valuable discussions with Dr J. Blackstock (UCL), and data processing and analysis performed by graduate student A. Vukandinovic at the University of Waterloo. We thank Dr D. McNeall (UKMO) for making his $R$ code publicly available at https://github.com/dougmcneall/famous-git. The Pacific Northwest National Laboratory is operated for the U.S. Department of Energy by Battelle Memorial Institute under contract DE-AC05-76RL01830.




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
