# Peer review of "Quantifying uncertainty from aerosol and atmospheric parameters and their impact on climate sensitivity"

_Atmospheric Chemistry and Physics, 2018_

## Referee Comment (RC1) · Anonymous Referee #2 · 31 Oct 2018

The study by Fletcher et al. provides an assessment of the role of compensation between uncertainties in aerosol forcing and atmospheric parameters, and their impact on climate sensitivity. From their analyses they conclude that structural uncertainties have a higher contribution to uncertainty in climate sensitivity than parametric uncertainties. This is a really interesting study. However, since the result presented are quite complex I have among others some suggestions on the presentation of the results that should be considered before publication in ACP.

**General comments:**
(1) I can follow your reasoning why you perform your analyses with such a coarse

resolution. However, I am not so convinced that it does not have any influence on your results. If you use a coarse resolution gases as well as aerosols will probably not as accurately simulated than using a higher resolution and thus I would expect an effect on radiative forcing. Can't you do a simple test? For example you could use the combination of the input parameters where you found in your analyses the highest uncertainty and repeat the analyses to check if the results are still the same when you apply model simulations with a higher resolution.

(2) The matrix plots are quite difficult to read. Wouldn't it be possible to use colours or frames to emphasize the correlations that are significant or have a high correlation coefficient? Further, the figures would be better readable if the labelling of input and output would be at the x and y-axis. I also would suggest to use different colours to separate between input and output.

**Specific comments:**
P1, L12: Shouldn't it be plural? Thus "uncertainties"?

P2, L2: There are also uncertainties concerning the emissions of trace gases. This should be mentioned here as well.

P4, L4: The abbreviation AGCMs has not been used yet.

P5, L3: Large scale transport is generally poorly simulated when model resolution is low. Of course, resolution does not solve all problems with the circulation problem, but it has been shown that it definitely looks better when a higher resolution is used.

P5, L1: O[$10^5$]? What is the O standing for?

P5, L1: If anyway an emulation is used, why is it then not possible to use a higher model resolution? How would a higher resolution change the computing time needed for the emulation?

P7, L30: I cannot see the correlation between x3 and AOD. Is these really correct? Is it not possible to improve the presentation of the results in the figures? See my specific comments on the figures below and above.

P8, L13: What is "Cess"? What do you mean with "Cess climate sensitivity"?

P9, Figure 1: The figure(s) are quite difficult to read. In the caption some guidance how to read this matrix plot should be given.

P10, L9: 10th percentile → 90% percentile (?)

P10, L14: Either use parenthesis around the reference Kay et al. (2014) or write "as in Kay et al. (2014)".

P10, L30: The abbreviation RMSE has not been introduced yet.

P11, Figure 2: There should be more space added between each row of figure panels. Wouldn't it possible to add a figure legend with the variable names or add them in the caption? Alternatively, this information could be given in a table. Additionally, the units at the axis should be given.

P12, Figure 3 caption: I would suggest to move the last sentence up, so that it is the second sentence in the caption.

P14, Figure 4 caption: by the → by
**Interactive comment**

P14, Figure 4 caption: Comma after "parameter" redundant?

P15, Figure 5: Also here I would suggest to add x und y-axis label for better readability or to give some guidance on how to read the figure in the figure caption. The read lines are not really helpful. Is there another way of presentation? Also here units should be given. Further, I would suggest to somehow mark (emphasize) in the figure the examples mentioned in the text.

P15, L1: Comma after "than" redundant?

P15, Figure 6: Similar comments on this figure. Add x and y axis labelling and chose another presentation than the red lines. These are rather confusing than helpful.

P19, Figure 7: Add colourbar so that the highest and lowest values (range of values) are easier to differentiate. Also here, the red lines are rather confusing than helpful.

P20, Figure 8: Same comments as for the previous figures.

P21, L26: Could you give here a number? How many percent?

---

## Referee Comment (RC2) · Anonymous Referee #3 · 13 Nov 2018

Review of manuscript entitled: ''Quantifying uncertainty from aerosol and atmospheric parameters and their impact on climate sensitivity" by Fletcher et al.

This work attempts analyzing the relative influence of aerosol and atmospheric parameters uncertainty on climate sensitivity. The methodology used here is to sample combinations of 9 parameters (4 representing the aerosols and 5 representing clouds) in the CAM4 general circulation model. The number of emulations to be performed is of the order of 10ˆ5. The design of the emulations and the methodology are well described but I believe that the conclusions of the paper are too far reaching. Here is why: Although the setup and the methodology appear to be well chosen, I would argue

that the authors sample only a small part of the aerosol space and not the most relevant parameters for ascertaining which part of the variation in climate sensitivity can be explained by atmospheric parameters and which one is due to aerosol parameters. Therefore part of the sentence included in the abstract and the conclusions: '' The atmospheric parameters explain more than 85% of the variance in climate sensitivity for the ranges of parameters explored here, with two parameters being the most important: one controlling low cloud amount, and one controlling the timescale for deep convection." can mislead readers. I recommend to edit this conclusion for the following reasons: 1) Page 4, line 18-19 the authors state: '' CAM4 does not include aerosol-cloud interactions, yet sulfate aerosols are known to be effective cloud condensation nuclei." This is right, sulfate aerosols are effective CCN (cloud condensation nuclei) as can be organic aerosols, seasalt and large dust particles. But since CAM4 does not include the effects of aerosol on cloud microphysics, trying to mimic it by varying the hygroscopicity (aptitude to uptake water) by sulphate is a shortcut that can hyde a large part of the effect aerosols have on cloud and precipitation. 2) The authors restrict the influence of aerosol parameters to studying the hygroscopicity of sulphate and the abundance of black carbon. To be complete, they should also integrate possible uncertainties of the distribution of organics, seasalt, dust and nitrate, their aging as well as their ability to be CCNs.

If these were taken into account, I doubt that the results of Figure 4 would remain the same as the aerosol influence on cloud microphysics would change the relative importance of aerosol versus atmospheric parameters on climate sensitivity. This is why I propose that the authors take out from their conclusion that: ''atmospheric parameters explain more than 85% of the variance in climate sensitivity". I anticipate that future publications will contradict this result. It is very likely that the authors are correct when they state that: ''low level clouds and the time scale for deep convection are the two most important parameters" since this will probably hold true even when the aerosol parameter space is better sampled.

With this caveat I believe that this study is of interest to ACP and that this paper could be published after the authors restrict one of their conclusions.

---

## Author Comment (AC1) · 28 Nov 2018

**Response to reviewer comments on "Quantifying uncertainty from aerosol and atmospheric parameters and their impact on climate sensitivity" by Fletcher et al. (2018).**

We thank the reviewers for taking the time to read and comment on our manuscript. The reviewer comments are reproduced here in blue, and our response to each comment is immediately below in black. The changes made to the text are described below each response, in italics.

**Response to Review Comments RC1:**

(1) I can follow your reasoning why you perform your analyses with such a coarse resolution. However, I am not so convinced that it does not have any influence on your results. If you use a coarse resolution gases as well as aerosols will probably not as accurately simulated than using a higher resolution and thus I would expect an effect on radiative forcing. Can't you do a simple test? For example you could use the combination of the input parameters where you found in your analyses the highest uncertainty and repeat the analyses to check if the results are still the same when you apply model simulations with a higher resolution.

We thank the reviewer for mentioning the potentially important effect of model horizontal resolution. This work was really a "proof-of-concept" study, and to minimize the overhead on our computing resources, we made the decision to focus on the low resolution (T31, approx. 3x4-deg) version of CAM4. The only real computing expense of this work is in producing the training simulations with CAM4; once the CAM4 simulations are complete, the emulator effectively runs for "free". The cost of running the training simulations with CAM4 at 2-degree (1-degree) resolution is a factor of 3 (15) higher than at T31, and our computing allocation in 2018 was insufficient to complete these simulations. However, we agree with the reviewer that an evaluation of the impact of resolution on parametric uncertainty is important, and we plan to carry out that study in future.

We considered the idea of running a test simulation at higher resolution with the optimal input parameters taken from the lower-resolution model. However, this may lead to erroneous conclusions, because the default values for the input parameters can actually be resolution-dependent (as mentioned on Page 7 of the manuscript). Therefore, we elect to hold off on testing the impact of resolution until we have sufficient computing resources to re-run the entire emulation procedure from scratch using higher resolution versions of CAM.

(2) The matrix plots are quite difficult to read. Wouldn't it be possible to use colours or frames to emphasize the correlations that are significant or have a high correlation coefficient? Further, the figures would be better readable if the labelling of input and output would be at the x and y-axis. I also would suggest to use different colours to separate between input and output.

We thank the reviewer for these suggestions, and we have attempted to implement them for all matrix-type plots in the revised version.

*Changes made:*

- *Background colours have been added to Fig.1 to emphasise the sign and magnitude of correlations. INPUT and OUTPUT labels have been added to make clear which panels are which.*

- *The clarity of axis labelling has been improved throughout*
- *Red lines have been replaced by dots to indicate the parameter settings of the default model*
- *Figs 6 and 7 have been combined for brevity, and the information made distinct.*
- *A colour bar has been added to the old Fig.8 (now Fig.7)*

Specific comments:

P1, L12: Shouldn't it be plural? Thus "uncertainties"?

Changed.

P2, L2: There are also uncertainties concerning the emissions of trace gases. This should be mentioned here as well.

We were not sure what the reviewer meant by "trace gases" here. In the next sentence, we already mention uncertainty in emissions of GHGs and aerosols.

P4, L4: The abbreviation AGCMs has not been used yet.

Changed to "atmospheric general circulation models".

P5, L3: Large scale transport is generally poorly simulated when model resolu- tion is low. Of course, resolution does not solve all problems with the circulation problem, but it has been shown that it definitely looks better when a higher resolution is used.

Thank you for this additional information. We have added a comment to reflect this, at the end of the sentence.

P5, L1: O[105]? What is the O standing for?

We were using "Big-O" notation to represent the order of the number of simulations that would be required to properly train the GCM. We have modified this to just say "$10^5$".

P5, L1: If anyway an emulation is used, why is it then not possible to use a higher model resolution? How would a higher resolution change the computing time needed for the emulation?

See detailed response about resolution above.

P7, L30: I cannot see the correlation between x3 and AOD. Is these really cor- rect? Is it not possible to improve the presentation of the results in the figures? See my specific comments on the figures below and above.

There is a fairly weak positive correlation between AOD and x3 (r ~ 0.2). This is clearer now that we have added background shading to the panels of Fig.1. The text has been amended to reflect this: *"AOD is … a weaker function of BC mass scaling"*.

P8, L13: What is "Cess"? What do you mean with "Cess climate sensitivity"?

This is a standard metric for estimating CS using atmospheric models with prescribed SST, and it is described in Section 2.1. In the revised manuscript, we have modified our notation to be consistent with that of Cess et al. (1989), where this method was first proposed, such that the Cess CS is denoted by the symbol \lambda. Page 8 is the first place where we present results using this quantity, and we wanted to emphasise to readers that it is the "Cess CS", rather than the equilibrium or transient climate sensitivity, that is being presented here.

P9, Figure 1: The figure(s) are quite difficult to read. In the caption some guid- ance how to read this matrix plot should be given.

See detailed response about figures above. But as suggested we have modified the caption to help the reader:

*The background shading of each panel indicates the strength and sign of the correlation between a particular pair of variables, with reds indicating positive, and blues indicating negative, correlations, and stronger correlations represented by darker shading. For example, there is a strong positive correlation between $x_5$ and FNET, and a weak negative correlation between $x_9$ and FNET.*

P10, L9: 10th percentile → 90% percentile (?)

The original article is correct: there are only ~10% of training cases that produce a value of CS lower than the default value.

P10, L14: Either use parenthesis around the reference Kay et al. (2014) or write "as in Kay et al. (2014)".

Wording is changed for clarity:

*Relative to modern ensembles with comprehensive ESMs \citep{kay_community_2014}, our sample of $n=350$ training cases provides a large ensemble of cases*

P10, L30: The abbreviation RMSE has not been introduced yet.

Definition is added at first use.

P11, Figure 2: There should be more space added between each row of figure panels. Wouldn't it possible to add a figure legend with the variable names or add them in the caption? Alternatively, this information could be given in a table. Additionally, the units at the axis should be given.

Figure has been changed:
- Spacing has been increased
- x-axis titles and units have been added

P12, Figure 3 caption: I would suggest to move the last sentence up, so that it is the second sentence in the caption.

Caption has been changed accordingly.

P14, Figure 4 caption: by the → by

Caption has been changed accordingly.

P14, Figure 4 caption: Comma after "parameter" redundant?

Matter of style. Comma is left as is.

P15, Figure 5: Also here I would suggest to add x und y-axis label for better readability or to give some guidance on how to read the figure in the figure caption. The read lines are not really helpful. Is there another way of presentation? Also here units should be given. Further, I would suggest to somehow mark (emphasize) in the figure the examples mentioned in the text.

Figure has been improved, as discussed above. We have added more description to the caption to make this figure easier to read:

*The blue shading in each panel indicates the density of plausible cases for each pair of input parameters ($x_1$ -- $x_9$). Plausible cases are defined as the subset of $n=100,000$ emulated samples with SS$>0.85$, and dAOD$<0.08$. Darker shading indicates higher density, and white areas indicate zero density (i.e., no plausible cases). The default values for each parameter in CAM4 are indicated by red dots in each panel (except for $x_4$, which has no default).*

P15, L1: Comma after "than" redundant?

Wording is changed for clarity:

*compressed toward a central value that is slightly higher than the default value in CAM4*

P15, Figure 6: Similar comments on this figure. Add x and y axis labelling and chose another presentation than the red lines. These are rather confusing than helpful.

P19, Figure 7: Add colourbar so that the highest and lowest values (range of values) are easier to differentiate. Also here, the red lines are rather confusing than helpful.

Figures 6 and 7 have been combined, with low (high) CS cases shown in blue (red).

P20, Figure 8: Same comments as for the previous figures.

Figure has been improved (discussed above).

P21, L26: Could you give here a number? How many percent?

We have attempted to quantify the "relatively small degree" to which a model could be moved along the Kiehl curve:

> *However, the relatively modest impact of parametric uncertainty means that a model could move along the curve by, perhaps, only 10-15 \% relative to the spread among CMIP5 models.*

**Response to Review Comments RC2:**

This work attempts analyzing the relative influence of aerosol and atmospheric pa- rameters uncertainty on climate sensitivity. The methodology used here is to sample combinations of 9 parameters (4 representing the aerosols and 5 representing clouds) in the CAM4 general circulation model. The number of emulations to be performed is of the order of $10^5$. The design of the emulations and the methodology are well described but I believe that the conclusions of the paper are too far reaching. Here is why: Although the setup and the methodology appear to be well chosen, I would argue that the authors sample only a small part of the aerosol space and not the most rel- evant parameters for ascertaining which part of the variation in climate sensitivity can be explained by atmospheric parameters and which one is due to aerosol parameters. Therefore part of the sentence included in the abstract and the conclusions: '' The atmospheric parameters explain more than 85% of the variance in climate sensitivity for the ranges of parameters explored here, with two parameters being the most im- portant: one controlling low cloud amount, and one controlling the timescale for deep convection." can mislead readers. I recommend to edit this conclusion for the following reasons: 1) Page 4, line 18-19 the authors state: '' CAM4 does not include aerosol- cloud interactions, yet sulfate aerosols are known to be effective cloud condensation nuclei." This is right, sulfate aerosols are effective CCN (cloud condensation nuclei) as can be organic aerosols, seasalt and large dust particles. But since CAM4 does not include the effects of aerosol on cloud microphysics, trying to mimic it by varying the hygroscopicity (aptitude to uptake water) by sulphate is a shortcut that can hyde a large part of the effect aerosols have on cloud and precipitation. 2) The authors restrict the influence of aerosol parameters to studying the hygroscopicity of sulphate and the abundance of black carbon. To be complete, they should also integrate possible un- certainties of the distribution of organics, seasalt, dust and nitrate, their aging as well as their ability to be CCNs.

If these were taken into account, I doubt that the results of Figure 4 would remain the same as the aerosol influence on cloud microphysics would change the relative impor- tance of aerosol versus atmospheric parameters on climate sensitivity. This is why I propose that the authors

take out from their conclusion that: ''atmospheric parameters explain more than 85% of the variance in climate sensitivity''. I anticipate that future publications will contradict this result. It is very likely that the authors are correct when they state that: ''low level clouds and the time scale for deep convection are the two most important parameters'' since this will probably hold true even when the aerosol parameter space is better sampled.

We thank the reviewer for this suggestion, and we agree with the comment and recommendation to amend the conclusions of the study to better represent what can be concluded from our experimental setup.

*Changes made:*

- Abstract*: we removed the mention of 85% variance explained, and added caveats about model configuration*

    *In this experimental setup where aerosols do not affect the properties of clouds, the atmospheric parameters explain the majority of variance in climate sensitivity, with two parameters being the most important:  one controlling low cloud amount, and one controlling the timescale for deep convection.  Although the aerosol parameters strongly affect aerosol optical depth, their impacts on climate sensitivity are substantially weaker than the impacts of the atmospheric parameters, but this result may depend on whether aerosol-cloud interactions are simulated.*

- Conclusions: We retain the 85% have added a new paragraph focusing on the limitations of not explicitly representing aerosol-cloud interactions in CAM4, or the role of other aerosol species.

    *However, while we attempt to quantify the impact of aerosol-cloud interactions (ACI) through the hygroscopicity parameter for sulfate aerosols $x_1$, the CAM4 model does not include direct simulation of ACI, which would be expected to substantially increase the importance of the aerosol parameters \citep{regayre_aerosol_2018}. Future work should quantify the importance of uncertainties in parameters related to subgrid-scale activation of cloud droplets by aerosols in newer versions of the CESM-CAM models that include these processes, and quantify their impacts on ACI and $\lambda$ \citep{golaz_cloud_2013}. In addition, our study focuses entirely on sulfate and black carbon aerosols, but important contributions to aerosol radiative forcing could be expected from uncertainties in the distribution of organic, sea salt, dust and nitrate aerosol, and the representation of their aging properties, and activation of cloud droplets \citep[e.g.,][]{chen_uncertainty_2006}. Therefore, while we expect the overall importance of $x_5$ and $x_9$ to be robust, we recommend caution in interpreting the precise numerical details of these results (for example, the 85 \% variance explained by atmospheric parameters), since these figures could be highly sensitive to the details of the model configuration.*

-